DoReMi: context-based prioritization of linear motif matches

Horn Heiko 1 heiko.horn@cpr.ku.dk
Haslam Niall 2 3
Jensen Lars Juhl 1
1 NNF Center for Protein Research, University of Copenhagen , Denmark
2 Complex and Adaptive Systems Laboratory, University College Dublin , Dublin , Ireland
3 Conway Institute of Biomolecular and Biomedical Science, University College Dublin , Dublin , Ireland
Dosztányi Zsuzsanna
Electronic publication date: 2014 Mar 20
Publication date: 2014
Volume: 2
Electronic Location ID: e315
Received 2013 Jul 30; Accepted 2014 Mar 1
Copyright: © 2014 Horn et al.
Copyright year: 2014
Copyright holder: Horn et al.
License: This is an open access article distributed under the terms of the Creative Commons Attribution License, which permits unrestricted use, distribution, and reproduction in any medium, provided the original author and source are credited.
License URL: https://creativecommons.org/licenses/by/3.0/

Keywords: Linear motifs, Web server, Prediction method, Protein interaction network

Funding: Novo Nordisk Foundation Center for Protein Research Science Foundation Ireland 08/IN.1/B1864 This work was in part funded by the Novo Nordisk Foundation Center for Protein Research. NJH is funded by Science Foundation Ireland (Grant 08/IN.1/B1864). The funders had no role in study design, data collection and analysis, decision to publish, or preparation of the manuscript.

==============================
Many protein domains bind to short peptide sequences, called linear motifs. Data on their sequence specificities is sparse, which is why biologists usually resort to basic pattern searches to identify new putative binding sites for experimental follow-up. Most motifs have poor specificity and prioritization of the matches is thus crucial when scanning a full proteome with a pattern.

Here we present a generic method to prioritize motif occurrence predictions by using cellular contextual information. We take 2 parameters as input: the motif occurrences and one or more of the interacting domains. The potential hits are ranked based on how strongly the context network associates them with a protein containing one of the specified domains, which leads to an increased predictive performance. The method is available through a web interface at doremi.jensenlab.org, which allows for an easy application of the method. We show that this approach leads to improved predictions of binding partners for PDZ domains and the SUMO binding domain. This is consistent with the earlier observation that coupling sequence motifs with network information improves kinase-specific substrate predictions.

Introduction

Physical interactions among proteins play a crucial role in cellular signaling. There are two types of binding: domain–domain and domain–motif. For the latter type, a conserved domain interacts with a short peptide sequence (linear motif), which may be subject to post-translational modification (PTM) (Linding, 2010). Many proteins carry more than one domain, which allows them to bind to multiple different proteins and thereby act, for example, as scaffolds. Moreover, recent work has shown interdependence of PTM sites both with each other and with linear motifs (Minguez et al., 2012). Studying these binding sites and the domains that recognize them can thus help decipher the complex signalling networks in the cell.

The computational task of dealing with motifs can be split into two separate tasks: the discovery of novel, not known motifs and the search for new instances of already known motifs. The discovery of novel motifs is challenging, nonetheless a number of techniques have been developed to address this problem (Chou & Schwartz, 2011; Davey et al., 2010; Lam et al., 2010). More recently structural approaches have been applied to the problem (Pathak et al., 2013) which, similar to other tools, rely on filtering out false positives in the discovery phase of motif identification. Physicochemical approaches have been developed to uncover patterns in the primary sequence that are characteristic of short peptide binding region (Dosztányi, Mészáros & Simon, 2009), and machine learning techniques employed to learn motifs from literature curated datasets (Lieber, Elemento & Tavazoie, 2010; Mooney et al., 2012; Nguyen Ba et al., 2012). Also, several resources exist that attempt to catalogue instances of linear motifs, incorporating manually curated and computational predictions of motifs (Dinkel et al., 2011; Mi et al., 2012; Sigrist et al., 2010). The discovery of both novel motifs and new instances is hindered by the problem of finding statistical significant matches. Many motifs are short peptide sequences and thereby appear in many different proteins by chance without representing a functional site (Obenauer, Cantley & Yaffe, 2003).

When searching for new instances, all occurrences of a motif should thereby be screened for their biological validity. Based on the protein sequence, predictors for protein folding, structure and disorder can be used to filter for potential motifs (Davey et al., 2011). As functionally important sections of proteins tend to be conserved, further improvement can be gained by incorporating orthology information. Additionally, keyword annotations can help to distinguish the interesting proteins for the investigated processes (Ramu, 2003).

Further improvements can be achieved by integrating the cellular context of the interactions. As we try to filter for binding-associated motifs, the information about protein–protein interactions between the substrates and a protein containing the binding domain is an obvious approach. As long as this information is available, the question of biological relevance can be coupled to the question of co-localization and association; a binding site that will never be in proximity to the corresponding domain is biological not relevant in the studied context.

Previous work has shown that contextual information in the form of known physical interactions (Mi et al., 2012) or functional associations (Linding et al., 2007; Weatheritt & Jehl, 2012) can improve motif based approaches. These methods are publicly available, but limit the query to a pre-defined set of motifs and interacting partners, which prevents their use for analysis of not already known motifs.

Here, we present DoReMi (Domain aided Regular Expression Mining), a tool that allows the user to specify any motif and the associated binding domain(s). By integrating the protein–protein association data into the scoring schema, we extend the usage from being a subsequently applied filter to a factor for ranking found potential instances. The algorithm is publicly available using the web interface at doremi.jensenlab.org.

Materials & Methods

The algorithm takes a motif representation as input and scans a selection of representative sequences to find matching instances. To filter these hits, the score of a motif search gets combined with the protein–protein association scores derived from the STRING database to rank the occurrences (Franceschini et al., 2013). Previous work has demonstrated the utility of allowing both regular expressions (RegEx) and position specific scoring matrices (PSSM) to generate motif definitions (Haslam & Shields, 2012); therefore we allow both of these as input as well as the result of any more advanced machine learning approach with a qualitative result.

The association scores are generated from the STRING confidence association scores. As the database only provides direct associations, indirect paths have to be calculated to cover also indirect interactions like e.g., scaffolding. Indirect interactions are computed by multiplying connecting edges on the path between the proteins of interest. To reduce the effect of over-connected nodes (hubs), we penalize for the connectivity by normalizing for the sum of all connecting edges of the node. The final score of an interaction can be described by the formula: ∏n=2nlastCnn−1→nn1+∑Cnn→nx−Cnn−1→nn+Cnn→nnβ

with Cnn−1→nn being the confidence score of two nodes interacting and ∑Cnn→nx being the sum of the confidence scores of all connecting edges of nn. The parameter β was set based on our experience in the NetworKIN project to a value of 0.34.

After extending the network to cover indirect paths, the dataset covers a wide range of associations between single proteins, each represented by a single score. This score can be interpreted as the likelihood of two proteins interacting with each other.

The potential interaction partners are recruited by selecting all proteins containing the interacting domain(s). To find these proteins, the Hidden-Markov-Model (HMM) representations provided by the PFAM database are used to scan the whole STRING sequence database for matching structures (Punta et al., 2012). For the scan, we use the HMMER3 package with default parameters as stated by PFAM. These domain-protein associations are pre-calculated to later increase the performance of the web service. All proteins carrying at least one of the interacting domain(s) of interest are treated as potential interactors. In the case of multiple potential interactors for one substrate, the highest scoring one is considered the most probable and its score is used in the subsequent calculations.

The result from the PSSM search and the derived contextual network score are combined using a weighting factor to be able to adjust the algorithm to different biological cases. Final Score=Motif1−α*Associationα.

This weighting factor is used to account for the different impact the two factors have in the biological context. For example, the SUMO binding domain has a strongly conserved motif, leading to the assumption that the motif is sufficient to achieve good predictive performance. Conversely, the process of ubiquitination is well known for not having any preference for certain amino acids surrounding the modified residue, thus it is favorable to put more weight on the context information. If the motif search method only returns a binary value (e.g., regular expression returns only matches but no scores), DoReMi sorts the matches by the context network score alone.

To make the algorithm easily accessible and applicable, we developed a web service implementing the described workflow. The interface of this webservice allows for multiple different inputs for the motif finding part, which are subsequently enhanced with the pre-computed network information.

Results and discussion

To estimate the performance of our method, we selected two sets of binding sites, which have been previously been used for training of SUMOylation predictors (Matic et al., 2010; Ren et al., 2009). The family of SUMO (Small Ubiquitin-like Modifier) proteins comprises of multiple small members that are covalently attached to the proteins. As the name suggests, the proteins are similar to ubiquitin, and the whole enzymatic cascade is alike to the process of ubiquitination. SUMOylation is involved in functions like protein stability, nuclear-cytosolic transport, and transcriptional regulation. The SUMO binding motif can be split into 2 groups: the first motif is a strongly conserved one whereas the second is less restrictive.

The older dataset from Ren et al. (2009) was used to calculate the PSSM and set the weighting factor α. We subsequently benchmarked the resulting predictor on the newer dataset by Matic et al., removing all sites that have been already used in the training. To compare the approach of creating a PSSM to a regular expression, we used the SUMOylation motif VILMAFPK.E

from the Eukaryotic Linear Motif (ELM) database (Dinkel et al., 2011).

In addition, we trained a predictor based on the combined datasets and benchmarked its performance by five-fold cross validation. The results of the partitioning were slightly better than the results from training on one dataset and benchmarking on the other. This suggests that cross validation gives a realistic performance estimate.

Given this, we also used cross validation to test the ability of DoReMi to predict binding partners for PDZ domains (Kim et al., 2012). PDZ domains are abundant in eukaryotes and eubacteria and commonly bind to the C-terminal end of proteins (Ponting, 1997). They are often associated with other domains like SH3 and therefor play an important role in the cell signaling (Zhang et al., 2011). The ELM database lists three different motifs, although they can be easily joined as they differ only in position four: …DE.ACVILF$…VLIFY.ACVILF$…ST.ACVILF$…DEFILSTVY.ACVILF$.

Overall, the motif is relatively loose and the C-terminal location may be the strongest indicator for a real occurrence.

The combined set for SUMOylation contained 1063 sites while the publication by Kim et al. (2012) provided 367 instances. For the benchmarking, we pre-defined α to 0.4 and 0.8 for the SUMO and PDZ test sets respectively. We selected these values based on the conservation of the known sequences, as one can expect a greater influence of the cellular context, given the higher flexibility of the motif. Cross validation was repeated five times and the reported numbers are the median performance.

When selecting an appropriate score cut off by fixing the false positive rate (0.3 for SUMO, 0.15 for PDZ) we can show an improvement of the true positive rate by up to 15 percentage points for the PDZ domain and 10 for the SUMO binding domain. The smaller increase for the SUMO domain is explained by the already good performance of the motif-based predictor, which leaves less room for improvement. These examples use simple motif predictors; for comparison, we also applied the context information to an already existing SUMO predictor (SUMOsp) to investigate the possibility of improving on more advanced methods. We acknowledge that the result will be influenced by the fact that this benchmark uses true positives on which the predictor got trained. The shown improvement by using contextual information should be valid. Comparable to the first examples, we see an improvement of the true positive rate by up to 10 percentage points, which motivated the inclusion of this possibility into the web interface. A more detailed listing of the performance numbers can be seen in Table 1. Although the overall performance is not significantly better for some of the examples, in a real world scenario, the method performs better. The stated improvements mean, that if we would have a list of results and sort it by the final score, true positives are enriched in the beginning of this list. This is important, as the search of new instances of known motifs is mainly dominated by the task of filtering out false positives.

Table 1 Detailed performance numbers for the benchmarking.

The first number states the performance without contextual information while the second number includes contextual information.

	Sensitivity	Specificity	AROC	
SUMO—Split set—RegExp	0.50/0.50	0.98/0.99	−/0.46	
SUMO—Split set—PSSM	0.59/0.62	0.96/0.96	0.82/0.84	
SUMO—Partitioning	0.75/0.77	0.90/0.90	0.82/0.84	
PDZ—Partitioning	0.43/0.49	0.90/0.90	0.70/0.72	

These results show that, although advanced methods like SUMOsp perform better than the basic approach of combining a RegEx or PSSM with contextual information, it can also be improved by the same approach. This shows the flexibility and versatility of this approach. Especially under the current state of lack of data that we see in the field of linear motif, it can be of great help in discriminating between true and false positive predictions. It also adds a non-sequence specific layer to many methods.

The web interface allows the user to easily apply the algorithm in a simple two-step process. The first step allows the user to define the potential binding sites. The first input type is specifying a regular expression; further details on the allowed format options are shown in Fig. 1. Alternatively, the user can provide a set of known binding peptides, which will be used to calculate a PSSM. Both the regular expression and the PSSM will be used to scan the human proteome for occurrences. As last option, the user can submit predictions from other motif search engines by providing identifier, position and score. These identifiers and positions have to be prior been mapped to the background sequence set used in DoReMi (STRING database version 9.05).

The second step allows for the selection of one or more potential binding domains. The interface allows the user to search for Pfam domains by their accession, id or description. From the list of search results, the user can select the relevant domains (Fig. 2A). If necessary, multiple searches can be performed to select differently named domains.

Figure 1 Flowchart of the typical workflow of DoReMi.

(1) The user typically provides the motif description in 3 different ways: regular Expression(s) define a motif by the basic format allowed in most implementations. Amino acids are defined by their one-letter code with “.” standing for any. Multiple potential residues can be encoded by using square brackets, e.g., “[DE]”. To quantify selected residues, we allow the basic operators “*”, “+”, “{2}” or “{3, 4}”. To use a PSSM for the motif search, the user simply provides a set of known binding motifs; these are used to calculate the amino acid distribution at each position, correcting for the overall amino acid distribution of the proteome. As a last option, users can provide results from other tools like SLiMsearch. (2) The second required input is the set of interacting domains. We provide the domains in PFAM-A. Proteins carrying any of the selected domains are defined as potential interaction partners. The highest scoring interacting protein for each motif instance is selected as potential binding partner. (3) The two scores are combined to rank each instance of the found motif.

Figure 2 Web interface of DoReMi.

(A) The interface allows the user to search for Pfam domains by their accession, id or description. From the list of search results, the user can select the relevant domains. If necessary, multiple searches can be performed to select differently named domains. (B) The output page shows a brief summary of the analysis. This includes plots of score distributions for each score (motif, network and combined score) to aid in the selection of an appropriate score cut-off for downstream analysis of the results after downloading.

The output page shows a brief summary of the analysis (Fig. 2B). This includes plots of score distributions for each score (motif, network and combined score) to aid in the selection of an appropriate score cut-off for downstream analysis of the results after downloading.

Conclusions

Our work focused on making the integration of contextual information to enhance the search for motif occurrences into an easy to execute task. We show that the addition of protein–protein interaction knowledge can improve the predictive performance of existing methods, even when doing it in a fully automated manner. This can be especially useful in the field of linear motifs; as for these, due to lack of data, advanced methods are not easy to develop. Further performance enhancements can potentially be reached when improving the domain associations in a more confined way.

For future enhancements, we plan to expand the usability of the method and web interface. Currently, only the human proteome is available; however, the method is in principle applicable to any fully sequenced organism. We thus consider expanding it to major eukaryotic model organisms. We are also working on integrating DoReMi with existing motif resources such as the ELM database.

Additional Information and Declarations

Competing Interests

Author Contributions

The authors declare no competing interests.

Heiko Horn conceived and designed the experiments, analyzed the data, wrote the paper, prepared figures and/or tables, reviewed drafts of the paper.

Niall Haslam conceived and designed the experiments, analyzed the data, wrote the paper, reviewed drafts of the paper.

Lars Juhl Jensen conceived and designed the experiments, wrote the paper, reviewed drafts of the paper.

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
