# Peer review of "DoReMi: context-based prioritization of linear motif matches"

_PeerJ, doi:10.7717/peerj.315_

## Round 0.1 · original submission · Major Revisions

Both reviewers found a few shortcomings of your submission, requesting clarifications and further testing of your method. You will need to carefully address these comments in your revision.

·

Basic reporting

The article is well written, and self-contained.

Experimental design

To my understanding the authors are presenting a tool to identify instances of linear motifs in the form of regular expressions or PSSMs along with their interacting domains, and enhance the specificity of the prediction by including network (interaction partners from STRING) and structural information (Pfam domains). The novelty of this method lies in the fact that one is able to use this approach to search for novel motifs (as opposed to only known ones, which is the case with iELM), and it is flexible, taking as input regular expressions or PSSMs. Thus it represents a relevant contribution to the field.

It would be useful if the authors could clarify a few points:
The ‘Materials & Methods’ section does not provide sufficient information to fully reproduce the method. For example the authors state that they penalize for the connectivity of hubs. They do not specifically state how they do that. Also the weighting factor is only described qualitatively, but if one was to reproduce the method they would need more specific information about how it is calculated. Also they do not state how they deal with more than one instance of a motif in the same sequence.

Furthermore, since the linear motif –domain interaction are binary and direct interactions, even in the event of scaffolding, it is not clear why the authors include the indirect paths as I would expect that to introduce significant noise. Also it is not clear how, multiplying the connecting edges on a path between proteins, represents the likelihood of these two proteins to interact with each other.

Finally, it has been well established that filtering the query sequences for only disordered regions, leads to significantly improved prediction results. The authors don’t seem to have taken that into account and it might improve their performance if they do. Since it is so well established, it should be included in the Web server as an option, should one wish to limit their hits to only those lying in disordered regions.

Validity of the findings

The public webserver is a very useful tool for the community. However, there are some issues that should be addressed to make the output more useful.
1) The output is mixed up using gene names and ensemble ids in the ‘name’ field, which will make it very difficult to process for any bioinformatician wanting to analyze the data. This needs to be made consistent or they authors can make a new column for the Ensembl id introduced there, in case the gene name is not known .
2) A lot of the Ensembl ids are obsolete or retired, making it difficult to find out what the genes are. The Ensembl database used should be updated, because this will be an issue for any person trying to analyze or integrate the results of this method with other ones.
3) Optional: The names/ids linked out to a page showing the gene or protein information so that one can click and easily find what it is.
4) Optional: In the page where one selects the domains, it would be useful to have a browse button or something showing all the available domains for search, in case someone doesn’t know exactly the Pfam domain name that they are looking for (e.g. if they are used to using SMART or another database, or they can’t find the domain they want they can look to see if the name is slightly different than what they are searching for)

Reviewer 2 ·

Basic reporting

* The introduction could be expanded a little to include some of the major approaches developed so far to address the problem discussed in this paper such as structure based prediction of motif binding, and machine learning approaches. Also, specific introduction to the two system this method was tested on would be welcome. E.g. What are SUMO and PDZ binding domains and what are their associated motifs?
* The methodology section is also rather sparse and does not describe the method in sufficient scientific rigor in order for a third party to reproduce the reported results. E.g. how were over connected hubs penalized? what is the default value for alpha? what is the alpha value set for the two test cases? What are the parameters for the HMM scan of STRING? what is the 'fixed false-positive-rate' selected? What are the sizes of the training/test sets?
* The results section too lacks details, Table 1 in particular. What are the two values reported in each cell? The authors mention there was an improvement in the performance against SUMO with CV vs. Split-set, the AUCs though are the same (both of them), is there reason the improvement in sensitivity better than the decrease in specificity? [more on CV below] What was the performance of SUMOsp to begin with?
* I would suggest including a flow-chart describing the algorithm rather than a screen-shot of the web-server, that would be much more informative.
* Table 2 is superfluous in my opinion. This is not a major result but rather common knowledge, it could be mentioned as text in the methods section.

Experimental design

* The addition of protein-protein interaction data to motif scanning in order to improve PTM predictions (such in the case of SUMO) is a sound idea, and indeed it would seem to improve predictions. However, when linear motif binding is used to facilitate the protein-protein interaction to begin with (such as with PDZ domains) I fear the apparent improvement presented here is not due to generalization but rather due to known data already in STRING. I.e. had this interaction wasn't already known in STRING the algorithm wouldn't be able to predict it any better than just based on the motif. The authors should attempt to prove/disprove this. This doesn't mean the method is not useful but would help understand where the improvement stems from.
* The biological logic behind including 'interaction scores' for proteins not directly interacting with the binding-domain can be explained for 'scaffolds' as mentioned by the authors. I.e. the binding domain interacts with scaffold X which also interacts with the substrate. Is there any logic in including paths of length >2?
* "This suggests that cross validation gives a realistic performance estimate". I object this statement. The authors previously mentioned the two datasets contain overlapping motifs, I assume these were removed before CV. Still, random CV can easily, artificially, improve performance by a favorable split of the data which is all but realistic. How many times was the 5-CV performed? is the reported value an avg.? was it ran only once?

Validity of the findings

* While there is a lot of merit in including PPI data and network information in linear motif ranking. Overall, I am not convinced by the claim that DoReMi, constitutes an improvement in ranking hits based on motif searches. I think the authors should more thoroughly analyze their results, and potentially include additional test cases, to substantiate their claim.
* Once the performance of the approach is more robustly assessed, the streamlined web-server would indeed be of great use to a broad crowd of biologists.

---

## Round 0.2 · accepted · Accept

The reviewers concerns have been adequately addressed.

·

Basic reporting

The authors have answered all my comments to my satisfaction.

Experimental design

The authors have answered all my comments to my satisfaction.

Validity of the findings

The authors have answered all my comments to my satisfaction.